# RADAR: Redundancy-Aware Diffusion for Multi-Agent Communication Structure Generation

**Zhen Zhang** [1]  **Wanjing Zhou** [1]  **Juncheng Li** [2]  **Hao Fei** [3]  **Jun Wen** [4]  **Wei Ji** [1]

## Abstract

Compared with individual agents, large language model based multi-agent systems have shown great capabilities consistently across diverse tasks, including code generation, mathematical reasoning, and planning, etc. Despite their impressive performance, the effectiveness and robustness of these systems heavily rely on their communication topology, which is often fixed or generated in a single step. This restricts fine-grained structural exploration and flexible composition, resulting in excessive token utilization on simple tasks while limiting capability on complicated tasks. To mitigate this challenge, we introduce RADAR, a redundancy-aware and query-adaptive generative framework that actively reduce communication overhead. Motivated by recent progress in conditional discrete graph diffusion models, we formulate communication topology design as a step-by-step generation process, guided by the effective size of the graph. Comprehensive experiments on six benchmarks demonstrate that RADAR consistently outperforms recent baselines, achieving higher accuracy, lower token consumption, and greater robustness across diverse scenarios. Our code and data are available at https://github.com/cszhangzhen/RADAR.

## 1. Introduction

Large Language Model (LLM) based agents have achieved great success across a wide spectrum of domains, including code generation (Zhang et al., 2024), question answering (Xu et al., 2024) and web navigation (Chae et al., 2025), etc. Beyond single-agent paradigms, prior work shows that multi-agent configurations, whether cooperatively (Zhuge et al., 2024; Zhang et al., 2025c) or competitively (Zhu et al., 2025; Wu et al., 2024b), can surpass the capabilities of individual agents (Du et al., 2023; Zhang et al., 2025b), highlighting the emergence of collective intelligence (Piatti et al., 2024; Chen et al., 2025). However, the emergence of this collective intelligence is primarily shaped by the design of the communication topology, which specifies agent connectivity and information flow (Zhang et al., 2025a; Hu et al., 2025). This highlights the crucial of effective collaboration graph design for multi-agent system performance, making it a key focus of research in this area.

Prior work on multi-agent systems has largely been constrained to static, hand-crafted configurations, such as chain structures that impose sequential execution (Wei et al., 2022), star topologies that enable centralized coordination (Jin et al., 2025), tree architectures that support hierarchical collaboration (Yao et al., 2023), and fully connected topologies that facilitate global communication. Although effective in specific scenarios, these collaboration architectures lack the flexibility required to generalize across diverse tasks. For instance, a simple arithmetic task might require a brief, linear exchange, whereas complex code generation often demands more elaborate collaborative structures. Therefore, applying a single fixed collaboration pattern across all tasks either incurs redundant communication overhead for simple problems or limits performance on more complicate ones (Zhang et al., 2025a; Jiang et al., 2025).

To enhance flexibility, more recent research has shifted towards automatically designing task-adaptive multi-agent systems (Zhang et al., 2025c; Li et al., 2025). These methods can be broadly classified into three categories: agentic profiling methods (Chen et al., 2024a; Yuan et al., 2025), which employs a coordinating agent to facilitate information sharing and environment adaptation; search-based models (Zhang et al., 2025a; Shang et al., 2025; Zhang et al., 2025d), which explores the design space of multi-agent configurations, and graph learning approaches (Zhuge et al., 2024; Zhang et al., 2025c; Wang et al., 2025), which learn inter-agent connectivity patterns to support task collaboration. Despite their diverse design paradigms, these approaches are constrained by fundamental limitations. Agentic profiling methods (Zhang et al., 2025d; Yuan et al., 2025) rely

[1]National Key Laboratory for Novel Software Technology, Nanjing University, China, [2]Zhejiang University, [3]University of Oxford, [4]Mohamed bin Zayed University of Artificial Intelligence. Correspondence to: Zhen Zhang <zhen_zhang@nju.edu.cn>, Wei Ji <weiji@nju.edu.cn>.

*Proceedings of the 43rd International Conference on Machine Learning*, Seoul, South Korea. PMLR 306, 2026. Copyright 2026 by the author(s).

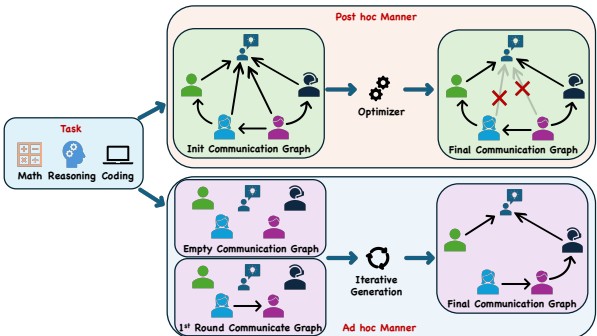

*Figure 1.* Comparison of workflows for designing communication topologies in LLM multi-agent systems. Compared with an ad hoc design approach, the post hoc paradigm might still incur redundant communication and fail to uncover novel structures.

heavily on a coordinating or meta-agent to profile roles, route information, or adjust collaboration strategies, which suffer from single-point bottlenecks. Search-based models (Shang et al., 2025; Zhang et al., 2025d) explore the design space of multi-agent configurations through heuristic or combinatorial search, which are often computationally expensive and poorly scalable. Graph learning approaches (Zhuge et al., 2024; Zhang et al., 2025c) explicitly learn inter-agent connectivity patterns, but they typically generate graphs in a single step conditioned on task descriptions, which restricts their ability to capture fine-grained and adaptive structures. These limitations highlight the need for more principled and flexible approaches that can dynamically generate collaboration topologies, efficiently adapt to diverse tasks, and balance structural expressiveness with communication efficiency.

At the same time, as multi-agent systems become more structurally elaborate, a complementary challenge emerges: their communication costs rise significantly due to overly complicated and often redundant collaboration designs. While rich interaction structures offer strong expressiveness, they can introduce unnecessary message passing, excessive token consumption, and coordination overhead, ultimately limiting scalability and efficiency. Empirical results in (Zhang et al., 2025b) demonstrate that more complex communication strategies require $2 \sim 11.8\times$ more tokens than simple chain topologies. To further improve collaboration efficiency, Zhang et al. (2025b) focus on pruning redundant connections in communication graphs, whereas Wang et al. (2025) propose to selectively dropping low-contributing agents. However, these methods operate on a predefined collection of agents and largely static interaction structures, where learning is limited to local modifications such as edge pruning or node dropping. Thus, they optimize efficiency in a post hoc manner, rather than jointly reasoning about structure formation and redundancy control during topology construction. As illustrated in Figure 1, they lack the capability to fundamentally redesign or generate collaboration topologies from scratch in a communication efficient man-

ner, limiting their expressiveness and adaptability to diverse or evolving task requirements.

To address the above challenges, we propose **R**edundancy-**A**ware **D**iffusion for Multi-**A**gent communication st**R**ucture generation (**RADAR**), which synthesizes the entire collaboration graph via iterative conditional graph diffusion models (Kong et al., 2023; Chen et al., 2023). Specifically, we incorporate the concept of effective size (Burt, 1992), which measures the non-redundant portion of a node's ego network, into the graph generation process to guide the construction of low-redundancy multi-agent communication structures. Meanwhile, this iterative, step-wise generation paradigm not only supports fine-grained exploration of the design space but also leverage the evolving partial graph at each step, enabling informed structural decisions and progressive mitigation of emerging redundancies. During inference, RADAR incorporates query-dependent contextual conditioning, thereby generating collective intelligence tailored to the specific task. Comprehensive evaluations are performed on six widely used benchmarks spanning reasoning, code generation and mathematical problem solving. The experimental results show that the proposed RADAR demonstrates substantial improvements compared with existing state-of-the-art baselines.

In summary, this work makes the following contributions:

- **Paradigm Redesign:** We develop an iterative, step-wise framework, which enables fine-grained generation of the multi-agent collaboration space, rather than relying on one-step generation.

- **Practical Framework:** The proposed framework automatically generates high-quality multi-agent collaboration systems and adaptively select efficient and high-performing solutions for queries of varying difficulty.

- **Experimental Evaluation:** Extensive experimental results across six public datasets show that our proposed model surpasses state-of-the-art baselines with different gains, while delivering improved token efficiency and enhanced robustness.

## 2. Related Works

**LLM-agent Collaboration.** The success of single agent systems (Shen et al., 2023; Wang et al., 2024a; Song et al., 2023) has inspired the development of multi-agent collaboration, giving rise to emergent collective intelligence. Numerous paradigms have been introduced to facilitate collaboration among multiple agents, including non-interactive query schemes, chain-of-thought prompting to debate mechanisms and fixed tree or star structures. Among them, LLM-Debate (Du et al., 2023) orchestrates multi-round debates in which agents independently articulate and refine their reasoning

before converging on a final answer. MetaGPT (Hong et al., 2024) formalizes collaboration by encoding standardized operating procedures into structured prompt sequences, yielding a sequential workflow. AutoGen (Wu et al., 2024a) supports flexible, conversational interaction patterns, including nested-chat, tree, and star topologies, enabling customizable coordination among agents. Nonetheless, their reliance on manual designs limits agents' capability to adapt to evolving scenarios.

**Multi-agents as Graphs.** Graphs provide a powerful and flexible framework for capturing interactions among diverse objects (Liu et al., 2024b; Zhang & He, 2025). Communication in multi-agents systems can naturally be represented as a graph, with nodes representing agents and edges encoding the flow of information. Early efforts primarily adopt complete or random graph structures (Qian et al., 2025), which assumes dense or arbitrary connection would be sufficient to facilitate effective information exchange among agents. To eliminate manual pipeline construction, more recent research has focused on automated topology learning. For instance, GPTSwarm (Zhuge et al., 2024) incorporates node-level optimization to adapt agent prompts and edge-level optimization to refine communication patterns. G-Designer (Zhang et al., 2025c) leverages a variational graph auto-encoder for encoding and decoding multi-agent interaction topologies. MaAS (Zhang et al., 2025a) learns a probabilistic, continuous architecture distribution from which task-adaptive multi-agent systems can be sampled for different queries. However, these approaches remain limited by their one-step generation mechanisms, which prevent them from exploring fine-grained collaboration structures and capturing the progressive dependencies inherent in complex multi-agent coordination.

**Graph Diffusion Models.** The rise of generative modeling has unlocked powerful mechanisms for graph synthesis, with conditional graph diffusion models achieving notable success in domains such as protein design (Yi et al., 2023), molecular generation (Liu et al., 2024a), materials (Klipfel et al., 2024) etc. Inspired by these advances, recent multi-agent systems research has begun to explore generating collaboration structures from scratch. For example, ARG-Designer (Li et al., 2025) constructs collaboration graphs using autoregressive modeling, while GTD (Jiang et al., 2025) employs conditional discrete graph diffusion for dynamic topology generation. Nevertheless, existing methods fail to explicitly reason about redundancy during structure formation, leading to unnecessary token overhead for simple tasks or limited effectiveness on more challenging problems. To overcome this limitation, we devise a novel redundancy-aware graph generation framework that conditions on effective topology size and constructs communication graphs step by step, enabling more adaptive and robust multi-agent collaboration.

## 3. Problem Formulation

In this section, we begin by establishing the notations and formalizing the key concepts from a topological perspective, followed by a formal definition of the LLM-based multi-agent communication protocol design.

### 3.1. Topological Structure

The multi-agent system can be naturally modeled as a directed graph $\mathcal{G} = (\mathcal{V}, \mathcal{E})$ with node set $\mathcal{V}$ and edge set $\mathcal{E}$. Each agent $v_i \in \mathcal{V}$ is formally defined as:

$$v_i = \{\texttt{Base}_i, \texttt{Role}_i, \texttt{State}_i, \texttt{Plugin}_i\}, \quad (1)$$

where $\texttt{Base}_i$ refers to the specific instance of a large language model; $\texttt{Role}_i$ denotes the functional role assigned to agent $i$; $\texttt{State}_i$ encapsulates the agent's accumulated knowledge and prior interactions; $\texttt{Plugin}_i$ specifies the external tools and plugins available to the agent, including web searcher, code compiler or file readers, etc (Wu et al., 2025; Wölflein et al., 2025). Upon receiving the prompt $\mathcal{P}_i$, each agent $v_i$ produces the corresponding response $\mathcal{R}_i$:

$$\mathcal{R}_i = v_i(\mathcal{P}_i) = v_i(\mathcal{P}_{\text{sys}}, \mathcal{P}_{\text{user}}), \quad (2)$$

where $\mathcal{P}_{\text{sys}} = \{\texttt{Role}_i, \texttt{State}_i\}$ indicates the system prompt, encapsulating the agent's role and internal state, while $\mathcal{P}_{\text{user}}$ corresponds to the user prompt, which incorporates task specifications, responses from other agents and externally retrieved knowledge.

### 3.2. Multi-Agent Communication Pipeline

Given the task query $\mathcal{Q}$, the multi-agent system conducts collaborative reasoning according to the collaboration graph $\mathcal{G}$, which specifies the pathways for information propagation among the agents. Unlike the conventional message passing schemes in graph neural networks (GNNs) (Kipf & Welling, 2017; Hamilton et al., 2017; Veličković et al., 2018), agent activations within each round follow an execution order derived from a topological sorting of the communication graph, ensuring that all prerequisite inputs are available before an agent is invoked. This collaborative procedure may be repeated for $K$ rounds to enable progressive refinement. Specifically, at round $k$, the mapping function $\phi(\cdot)$ determines the execution index of each agent:

$$\phi : \mathcal{G} \mapsto \sigma, \sigma = [v_{\sigma_1}, v_{\sigma_2}, \cdots, v_{\sigma_N}], \\ \text{s.t. } \forall i > j, \ v_{\sigma_i} \notin \mathcal{N}_i(v_{\sigma_j}), \quad (3)$$

where $\sigma$ specifies the agent execution sequence. $\mathcal{N}_i(v_{\sigma_j})$ denotes the in-neighborhood of agent $v_{\sigma_j}$. The execution order guarantees that an agent $v_{\sigma_i}$ is activated only after all agents from which it receives information have complete their execution. Once the execution order is established,

each agent generates its response as follows:

$$\mathcal{P}_i^k = \{\mathcal{P}_{\text{sys}}^k, \{\mathcal{Q}, \cup_{v_j \in \mathcal{N}_i(v_i)} \mathcal{R}_j^k\}\}, \ \mathcal{R}_i^k = v_i(\mathcal{P}_i^k), \quad (4)$$

where the response $\mathcal{R}_i^k$ is conditioned on the system prompt $\mathcal{P}_{\text{sys}}^t$ together with a context prompt composed of the query $\mathcal{Q}$ and the incoming messages from neighboring agents. Upon completion of $K$ round communications, an aggregation function is employed to consolidate these responses into the final solution as follows:

$$\mathcal{S}^K \leftarrow \texttt{Aggregate}(\mathcal{R}_1^K, \mathcal{R}_2^K, \cdots, \mathcal{R}_N^K). \quad (5)$$

The `Aggregate` function can be implemented flexibly and might take various forms, such as majority voting (Chen et al., 2024b; Zhuge et al., 2024), concatenating and consolidating responses from all agents for final decision making (Zhang et al., 2025b; Jiang et al., 2023), or directly adopting the output of the last agent $\mathcal{R}_{\sigma_N}^K$ (Qian et al., 2025; Zhang et al., 2025c; Li et al., 2025). The communication process may proceed for a fixed number of rounds $K$ or terminate adaptively based on an early-stopping criterion.

### 3.3. Automatic Multi-Agent Topology Design

For a given query $\mathcal{Q}$, a large language model based multi-agent system $\mathcal{G} = (\mathcal{V}, \mathcal{E})$ is automatically constructed to satisfy the following design objectives: *1. Effectiveness*: maximize the task utility, $u(\mathcal{G}(\mathcal{Q}))$; *2. Cost Efficiency*: minimize the financial cost, $c(\mathcal{G}; \mathcal{Q})$; *3. Adaptiveness*: adjust the topology in response to varying tasks. The system's objectives can be simultaneously represented in the following optimization framework:

$$\min_{\mathcal{G} \in \mathbb{G}} \mathcal{L}(\mathcal{G}; \mathcal{Q}) = -u(\mathcal{G}(\mathcal{Q})) + \alpha \cdot c(\mathcal{G}; \mathcal{Q}), \quad (6)$$

where function $u(\cdot)$ quantifies task-specific utility (e.g., accuracy, pass@1, etc.), $c(\cdot)$ measures the communication cost, and $\alpha$ is a trade-off hyper-parameter.

## 4. The Proposed RADAR Model

An overview of the proposed RADAR framework is illustrated in Figure 2, which consists of four principal stages. The workflow starts with task-relevant inputs, including the task query, the set of candidate agents, and available tools, etc. Based on these inputs, multiple baseline topologies are constructed to form a foundational dataset that captures the relationship between communication structures and task performance. This dataset is subsequently used to train graph diffusion models capable of generating high-performing graph topologies. Given a new task, the denoising network progressively refines an initially empty graph through iterative denoising, ultimately synthesizing a task-adaptive communication topology.

### 4.1. Effective Size as a Measure of Redundancy

Effective size is a classical graph theory concept that quantifies structural redundancy within a node's local neighborhood (Burt, 1992). It characterizes the extent to which a node's connections provide non-overlapping access, discounting neighbors that are highly interconnected and therefore likely lead to redundant information. Inspired by this concept, we adapt effective size to the setting of graph-based multi-agent collaboration, where nodes represent agents and directed edges encode information flow. In this context, we define the effective size of a node's incoming ego network as follows:

$$\varphi^i(v_k) = |\mathcal{N}_i(v_k)| - \frac{\sum_{j,q \in \mathcal{N}_i(v_k)} A_{jq} \mathbb{I}[r(j) = r(q)]}{|\mathcal{N}_i(v_k)|}, \quad (7)$$

where $\mathcal{N}_i(v_k)$ represents the in-neighborhood of agent $v_k$, while $A_{jq} \in \{0, 1\}$ encodes the information flow relationship between agents $v_j$ and $v_q$. The role function $r(\cdot)$ specifies the functional role of each agent, and $\mathbb{I}[\cdot]$ is an indicator function. This equation measures how many distinct and complementary information sources an agent effectively receives. A high incoming effective size indicates that agent $v_k$ benefits from diverse perspectives and complementary expertise, *i.e.*, low redundancy. Similarly, the outgoing effective size of agent $v_k$, i.e., $\varphi^o(v_k)$, measures redundancy in information dissemination. An agent is structurally effective when it distributes information across diverse, non-overlapping execution paths, while maintaining sufficient information diversity. Together with the incoming effective size, these two metrics provide a principled characterization of local information efficiency in directed multi-agent collaboration graphs. Formally, we define the effective size for agent $v_k$ in directed communication graphs as follows:

$$\varphi(v_k) = \varphi^i(v_k) \cdot (1 - \beta) + \varphi^o(v_k) \cdot \beta, \quad (8)$$

where $\beta \in [0, 1]$ is a trade-off hyper-parameter that balances the contributions of incoming and outgoing effective size. The value of $\varphi(v_k)$ will play an important role in the communication graph generation process, guiding the model in reasoning over structure information and controlling redundancy.

### 4.2. Redundancy-Aware Graph Diffusion

We formulate the multi-agent communication topology as a directed graph $\mathcal{G} = (\mathcal{V}, \mathcal{E})$ characterized by a binary adjacency matrix $\mathbf{A} \in \{0, 1\}^{N \times N}$, where $A_{ij} = 1$ indicates the presence of an edge $e_{ij} \in \mathcal{E}$ and $A_{ij} = 0$ otherwise. Each edge $e_{ij}$ encodes the directed information flow from agent $v_i$ to agent $v_j$. Under this formulation, the design of multi-agent communication structure is formulated as a graph generation problem, and our goal is to learn a redundancy-aware graph generative model from a collection of training communication graphs via diffusion model.

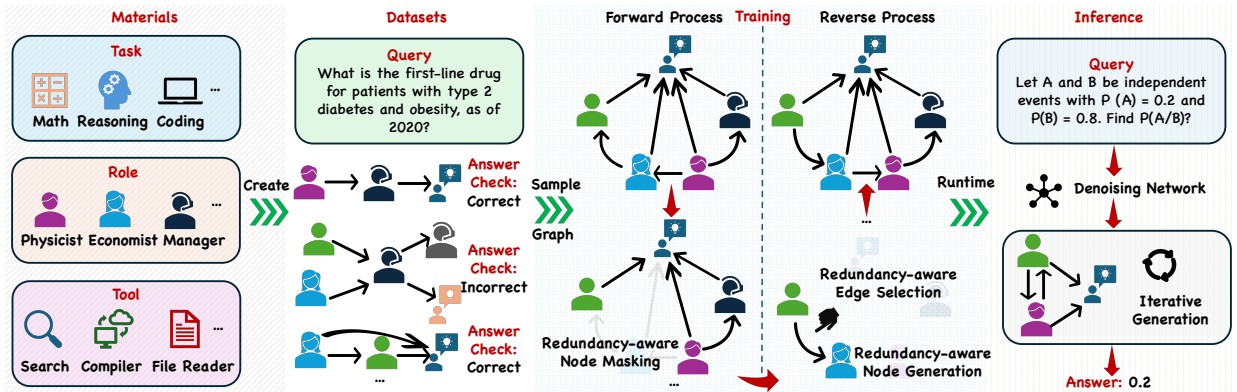

*Figure 2.* Overview of the RADAR framework. It starts from task-specific inputs and constructs baseline topologies to train a graph diffusion model. The trained denoising network then iteratively refines an initially empty graph to synthesize a task-adaptive topology.

**The Forward Process.** Motivated by the recent advances in discrete graph diffusion models (Kong et al., 2023; Chen et al., 2023; Yang et al., 2023), we introduce a forward diffusion process that progressively masks nodes together with their connected edges. To guide this process, we design a redundancy-aware ordering network that prioritizes nodes based on their effective size, thereby imposing structured regularities that simplify generative learning. Intuitively, graphs with high effective size tend to decompose into weakly overlapping substructures, making it more tractable to be generated incrementally. Accordingly, we employ an ordering network $q_\psi(\pi|\mathcal{G}_0, \varphi)$ that samples a node $v_{\pi(t)}$ to be masked at each diffusion step $t$, yielding the corresponding partially masked graph $\mathcal{G}_t$. Under this formulation, the ordering network operates in a recurrent manner across diffusion steps as follows:

$$q_\psi(\pi|\mathcal{G}_0, \varphi) = \prod_t q_\psi(\pi_t|\mathcal{G}_0, \varphi, \pi_{(<t)}). \quad (9)$$

Specifically, at diffusion step $t$, the selection of the $t$-th node $\pi_t$ is modeled as a conditional distribution that depends on the original graph $\mathcal{G}_0$, the node-level effective size $\varphi(v_t)$, and the previously sampled ordering $\pi_{(<t)}$. The graph structure is subsequently processed by a graph neural network (GNN) (Kipf & Welling, 2017; Hamilton et al., 2017; Zhang et al., 2021) to obtain node-level representations. To explicitly capture the partial orderings, positional encodings (Vaswani et al., 2017) are incorporated into the node features prior to message passing. Let $h_t$ denote the resulting embedding of node $v_t$ produced by GNN. The conditional distribution $q_\psi(\pi_t|\mathcal{G}_0, \varphi, \pi_{(<t)})$ is then parameterized as a categorical distribution over nodes:

$$q_\psi(\pi_t|\mathcal{G}_0, \varphi, \pi_{(<t)}) = \frac{\exp(h_t + \varphi(v_t))}{\sum_{j \notin \pi_{(<t)}} \exp(h_j)}, \quad (10)$$

where the output is a scalar score for candidate node $v_t$, representing the probability of being selected at step $t$. Through this formulation, the ordering network could sequentially select nodes in a structure-aware and redundancy-informed manner.

**The Reverse Process.** During the reverse generative phase, a denoising network $p_\theta(\mathcal{G}_t|\mathcal{G}_{t+1}, \mathcal{Q})$ progressively reconstructs the graph by inverting the forward diffusion process conditioning on the query $\mathcal{Q}$, thereby enabling the generated topology to be task-adaptive. At step $t$, the denoising network takes the partially masked graph $\mathcal{G}_{t+1}$ as input and maps each node $v_i$ into a latent embedding space. The node representations are then iteratively refined through graph neural network message-passing operations. Specifically, at the $(l+1)$-th message passing layer, the embedding of node $v_i$ is updated through aggregating attention-weighted messages from its incoming neighbors:

$$\alpha_{i,j} = \frac{\exp(\text{ReLU}(\mathbf{a}^\top[\mathbf{W}\mathbf{h}_i^l||\mathbf{W}\mathbf{h}_j^l]))}{\sum_{k \in \mathcal{N}_i} \exp(\text{ReLU}(\mathbf{a}^\top[\mathbf{W}\mathbf{h}_i^l||\mathbf{W}\mathbf{h}_k^l]))}, \quad (11)$$

$$\mathbf{h}_i^{l+1} = \text{ReLU}(\sum_{j \in \mathcal{N}_i} \alpha_{i,j}\mathbf{W}\mathbf{h}_j^l). \quad (12)$$

Here, $\mathbf{W}$ denotes the learnable weight matrix and $\mathbf{a}$ represents the attention vector. $\mathcal{N}_i$ is the neighborhood of node $v_i$, while ReLU refers to the activation function. After passing message $L$ rounds, the denoising network produces the final embedding $\mathbf{h}_i^L$ for each node. We then apply a bias term based on the effective size, updating the representation as $\mathbf{h}_i^L = \mathbf{h}_i^L + \varphi(v_i)\mathbf{1}$. Based on these embeddings, multi-layer perceptions are utilized to predict the agent role of the newly recovered node $v_{\pi_t}$ and its connectivity to the set of previously denoised nodes $\{v_{\pi(>t)}\}$. Rather than generating edges in an autoregressive manner, the connections between $v_{\pi_t}$ and all existing nodes are inferred jointly using a mixture of multinomial distributions. This design captures dependencies among edge variables while reducing the number of generation steps to $\mathcal{O}(N)$, which enables efficient graph structure generation.

## 4.3. Training Objective

To optimize the model's parameters, we use a reinforcement learning based training strategy to jointly update the diffusion ordering network $q_\psi(\pi|\mathcal{G}_0, \varphi)$ and the denoising network $p_\theta(\mathcal{G}_t|\mathcal{G}_{t+1}, \mathcal{Q})$ using gradient descent. At each training iteration, for the $i$-th training graph $\mathcal{G}_0^{(i)}$, we generate $M$ diffusion trajectories by sampling node-masking orderings $\pi^{i,m}$ from the ordering network. Each trajectory consists of a sequence of partially masked graphs $\{\mathcal{G}_t^{i,m}\}_{1 \le t \le N}$, where $N$ denotes the number of nodes. For each trajectory, diffusion steps are executed over $T$ time steps. Conditioned on these sampled trajectories, the denoising network is trained to minimize the negative variational lower bound using stochastic gradient descent. For notation clarity, the superscript $i$ is omitted in the following formulation:

$$\nabla_\theta \mathcal{G} = \sum_{m,t} \sum_{k \in \pi(\le t)} w_k^m \nabla \log p_\theta(\mathcal{G}_{v_k}^{\pi(>t)}|\mathcal{G}_{t+1}^m, \mathcal{Q}), \quad (13)$$

where $w_k^m = q_\psi(\pi_t^m = k|\mathcal{G}_0, \varphi, \pi_{(<t)}^m)$ denotes the probability assigned by ordering network to selecting node $v_k$ at diffusion step $t$. The term $p_\theta(\mathcal{G}_{v_k}^{\pi(>t)}|\mathcal{G}_{t+1}^m, \mathcal{Q})$ represents the denoising network's conditional distribution for jointly generating node $v_k$ and its connections to previously reconstructed nodes. The resulting weighted log-likelihood gradient specifies how the denoising model updates its parameters to favor the generation of structurally coherent nodes and edges while adhering to the reverse order of the diffusion process.

For the diffusion ordering network, it is optimized using a standard reinforcement learning technique, i.e., the REINFORCE algorithm (Williams, 1992), since the node masking orderings it produces are discrete and non-differentiable:

$$\nabla_\psi \mathcal{G} = \sum_m R^m \nabla \log q_\psi(\pi|\mathcal{G}_0^m, \varphi), \quad (14)$$

where $R^m$ provides a learning signal for the ordering network by measuring how well a sampled node-masking sequence supports high-likelihood reconstruction in the denoising model:

$$R^m = -\sum_t \sum_{k \in \pi(\le t)} w_k^m \log p_\theta(\mathcal{G}_{v_k}^{\pi(>t)}|\mathcal{G}_{t+1}^m, \mathcal{Q}). \quad (15)$$

In addition to structural fidelity, we explicitly optimize for downstream task utility. However, the utility function $u(\cdot)$ is often non-differentiable and computationally intractable, as it often relies on external API evaluations or black-box task executions (Li et al., 2023). To this end, we adopt a policy gradient based estimator to optimize Equation (6) as follows:

$$\nabla_\theta \mathbb{E}[\mathcal{G}] \approx \frac{1}{\mathcal{B}} \sum_{k=1}^{\mathcal{B}} u(\mathcal{G}^{(k)}(\mathcal{Q})) \nabla_\theta \log p_\theta(\mathcal{G}^{(k)}|\mathcal{Q}). \quad (16)$$

This formulation enables the model to directly leverage task-level feedback to guide topology generation, thereby encouraging agent communication structures that are aligned with task performance objectives. In practice, we adopt a periodic and subsampled utility evaluation strategy. The task utility is computed only for a subset of generated graphs at a fixed update frequency across training epochs and mini-batches. Through the joint updating of its key components, the proposed model enables the fully automated generation of multi-agent collaboration structures.

## 5. Experiments

### 5.1. Datasets

To comprehensively evaluate our proposed model, we conduct experiments across a diverse set of datasets including MMLU (Hendrycks et al., 2021) (*general reasoning*), GSM8K (Cobbe et al., 2021), MultiArith (Roy & Roth, 2015), SVAMP (Patel et al., 2021), AQuA (Ling et al., 2017) (*mathematical problem solving*), and HumanEval (Chen et al., 2021) (*code generation*). The dataset statistics and evaluation metrics are presented in Table 6 in the Appendix A.

### 5.2. Baselines

We compare RADAR against three types of baselines: (1) **Single Agent Methods** including CoT (Wei et al., 2022), ComplexCoT (Fu et al., 2023), SC (Wang et al., 2023); (2) **Multi-Agent Systems** including MultiPersona (Wang et al., 2024b), LLM-Debate (Du et al., 2023), LLM-Blender (Jiang et al., 2023), DyLAN (Liu et al., 2024c), AgentVerse (Chen et al., 2024c), MacNet (Qian et al., 2025); (3) **Autonomous Multi-Agent Systems** including AutoAgents (Chen et al., 2024a), GPTSwarm (Zhuge et al., 2024), ADAS (Hu et al., 2025), AgentSquare (Shang et al., 2025), AFlow (Zhang et al., 2025d), G-Designer (Zhang et al., 2025c), Agent-Prune (Zhang et al., 2025b), GTD (Jiang et al., 2025), MaAS (Zhang et al., 2025a), ARG-Designer (Li et al., 2025). Additional details are provided in Appendix B.

### 5.3. Implementation Details

We interact with GPT models through the OpenAI API[1], with experiments primarily conducted on `gpt-4o-mini`. For all runs, the temperature is fixed at $0.2$, and the maximum token budget per execution is set to $1,000$. Query representations are encoded using the `all-MiniLM-L6-v2` model (Wang et al., 2020), which outputs embeddings of dimension $384$. Across all datasets, we sample $50$ queries for model training. The initial diffusion dataset is constructed by instantiating a diverse set of baseline com-

---

[1]https://api.openai.com/v1/chat/completions

munication topologies (e.g., `fully connected`, `mesh`, `star`, `layered`, and `random`) with varying numbers of agents (e.g., 3 or 4 agents). For each configuration, we randomly assign agent roles and evaluate the resulting multi-agent system on sampled training instances (e.g., whether the final answer is correct or incorrect). This provides a diverse set of graph-structured samples to initialize the diffusion model. The ordering network is implemented as a three-layer RGCN (Schlichtkrull et al., 2018), while the denoising network adopts a three-layer GAT (Veličković et al., 2018) with hidden dimension setting as 32. The parameters are optimized using Adam (Kingma, 2014) with learning rates of $5 \times 10^{-4}$ for the diffusion-based ordering network and $1 \times 10^{-4}$ for the denoising network. The hyperparameters $\alpha$ and $\beta$ are tuned via grid search over the interval $[0, 1]$.

## 5.4. Performance Analysis

The experimental results reported in Table 1 demonstrate that our proposed RADAR is highly effective at designing multi-agent collaboration topologies. Specifically, RADAR consistently achieves the strongest performance across the six benchmarks. In particular, the multi-agent systems generated by RADAR outperform the vanilla single-agent baseline by margins ranging from $1.96\%$ to $6.59\%$. When compared with the recent strongest learning-based baselines, such as ARG-Designer, RADAR delivers an average performance improvement of $1.75\%$. Importantly, our results indicate that multi-agent collaboration does not always guarantee superior performance: several existing multi-agent topologies fail to consistently outperform single-agent systems like GPTSwarm and ADAS. This is largely because poorly coordinated agent interactions can introduce redundant reasoning, conflicting intermediate conclusions, or ineffective information aggregation, which might offset the potential benefits of collaboration. On the other hand, some existing multi-agent methods achieve strong empirical performance but suffer from substantial token consumption overhead, as shown in Figure 3. In contrast, RADAR attains superior accuracy with lower computational cost, illustrating a strong balance between effectiveness and token consumption.

Furthermore, we observe that the effectiveness of different systems varies substantially across evaluation scenarios with different levels of task difficulty. On relatively simpler reasoning benchmarks such as GSM8K, most baselines exhibit significant performance improvements over the single-agent setting, indicating that even straightforward forms of collaboration can be beneficial in low-complexity tasks. However, this performance gap narrows considerably in more challenging scenarios. For instance, in the most difficult code generation tasks, the majority of baseline methods achieve only modest gains of approximately $3\%$, suggesting limited

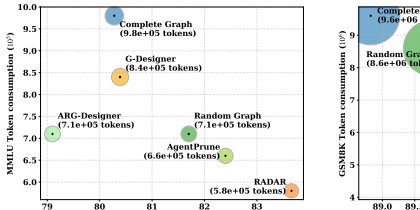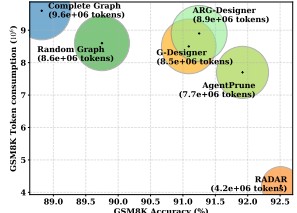

*Figure 3.* Visualization of the performance and token consumption.

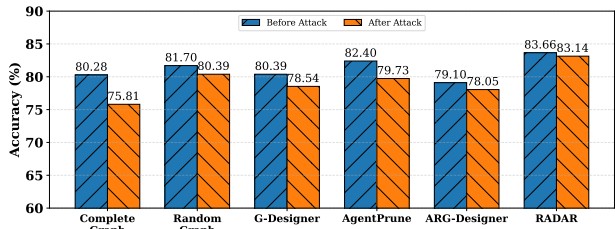

*Figure 4.* Before and after prompt attacks on MMLU dataset.

capability of their collaboration strategies under increased task complexity. In contrast, RADAR continues to deliver consistent benefits, achieving a $4.2\%$ improvement in this setting. These results highlight RADAR's superior ability to construct effective collaboration topologies that remain robust as task difficulty increases, further underscoring its advantage over existing multi-agent approaches.

## 5.5. Token Economical

One key advantage of RADAR lies in its capability to generate task-specific collaboration topologies, thereby avoiding unnecessary complexity and reducing token consumption. Figure 3 illustrates the trade-off between model performance and token usage. Our proposed RADAR demonstrates a favorable trade-off, maintaining strong performance while exhibiting high token efficiency. On the GSM8K dataset, RADAR is the most token-efficient approach, consuming only $4.2 \times 10^6$ tokens (including both prompt tokens and completion tokens), approximately half the token cost of GDesigner, while attaining the highest performance. This advantage becomes more pronounced as the number of evaluation samples increases, reflecting RADAR's task-adaptive design. In contrast, more complex communication structures, such as fully connected graphs, incur substantially higher token costs. By explicitly accounting for redundancy and exploring a fine-grained topology, RADAR attains superior token economy compared to existing approaches.

## 5.6. Robustness Analysis

Following prior work (Zhuge et al., 2024; Zhang et al., 2025c), we inject system prompt attacks into two of the five agents in the collaboration framework. More specifically, we compromise the role prompt of selected agents by alter-

*Table 1.* Performance comparison across three categories of baselines, including single-agent methods, multi-agent systems, and autonomous multi-agent systems. The best results are highlighted in bold. All methods, except those in the single-agent category, employ **five** `gpt-4o-mini`-based agents. The "Auto" column indicates whether a method supports autonomous topology design.

| Methods | Auto | MMLU | GSM8K | MultiArith | SVAMP | AQuA | HumanEval | Avg. |
|---|---|---|---|---|---|---|---|---|
| Vanilla | ✗ | 78.54 | 87.45 | 96.85 | 86.67 | 78.92 | 87.08 | 85.92 |
| CoT | ✗ | $79.26_{\uparrow0.72}$ | $87.10_{\downarrow0.35}$ | $96.31_{\downarrow0.54}$ | $87.33_{\uparrow0.66}$ | $75.20_{\downarrow3.72}$ | $88.13_{\uparrow1.05}$ | 85.55 |
| ComplexCoT | ✗ | $79.80_{\uparrow1.26}$ | $86.89_{\downarrow0.56}$ | $96.70_{\downarrow0.15}$ | $87.67_{\uparrow1.00}$ | $75.59_{\downarrow3.33}$ | $87.49_{\uparrow0.41}$ | 85.69 |
| SC (COT×5) | ✗ | $80.66_{\uparrow2.12}$ | $87.57_{\uparrow0.12}$ | $96.58_{\downarrow0.27}$ | $88.00_{\uparrow1.33}$ | $82.28_{\uparrow3.36}$ | $88.60_{\uparrow1.52}$ | 87.28 |
| MultiPersona | ✗ | $77.69_{\downarrow0.85}$ | $87.50_{\uparrow0.05}$ | $97.49_{\uparrow0.64}$ | $87.00_{\uparrow0.33}$ | $79.23_{\uparrow0.31}$ | $88.32_{\uparrow1.24}$ | 86.21 |
| LLM-Debate | ✗ | $80.56_{\uparrow2.02}$ | $89.47_{\uparrow2.02}$ | $97.33_{\uparrow0.48}$ | $89.00_{\uparrow2.33}$ | $79.70_{\uparrow0.78}$ | $88.68_{\uparrow1.60}$ | 87.46 |
| LLM-Blender | ✗ | $80.29_{\uparrow1.75}$ | $88.35_{\uparrow0.90}$ | $97.29_{\uparrow0.44}$ | $87.33_{\uparrow0.66}$ | $78.99_{\uparrow0.07}$ | $88.80_{\uparrow1.72}$ | 86.84 |
| DyLAN | ✗ | $79.86_{\uparrow1.32}$ | $89.98_{\uparrow2.53}$ | $97.12_{\uparrow0.27}$ | $88.67_{\uparrow2.00}$ | $79.59_{\uparrow0.67}$ | $90.42_{\uparrow3.34}$ | 87.61 |
| AgentVerse | ✗ | $78.39_{\downarrow0.15}$ | $89.91_{\uparrow2.46}$ | $97.50_{\uparrow0.65}$ | $88.33_{\uparrow1.66}$ | $77.47_{\downarrow1.45}$ | $89.29_{\uparrow2.21}$ | 86.82 |
| MacNet | ✗ | $79.55_{\uparrow1.01}$ | $87.95_{\uparrow0.50}$ | $96.03_{\downarrow0.82}$ | $86.00_{\downarrow0.67}$ | $79.23_{\uparrow0.31}$ | $84.57_{\downarrow2.51}$ | 85.55 |
| AutoAgents | ✓ | $79.59_{\uparrow1.05}$ | $87.69_{\uparrow0.24}$ | $96.42_{\downarrow0.43}$ | $86.34_{\downarrow0.33}$ | $78.65_{\downarrow0.27}$ | $87.64_{\uparrow0.56}$ | 86.05 |
| GPTSwarm | ✓ | $78.36_{\downarrow0.18}$ | $89.14_{\uparrow1.69}$ | $96.79_{\downarrow0.06}$ | $88.67_{\uparrow2.00}$ | $80.71_{\uparrow1.79}$ | $89.32_{\uparrow2.24}$ | 87.17 |
| ADAS | ✓ | $78.39_{\downarrow0.15}$ | $86.12_{\downarrow1.33}$ | $96.02_{\downarrow0.83}$ | $86.33_{\downarrow0.34}$ | $77.71_{\downarrow1.21}$ | $84.19_{\downarrow2.89}$ | 84.79 |
| AgentSquare | ✓ | $79.58_{\uparrow1.04}$ | $87.62_{\uparrow0.17}$ | $97.77_{\uparrow0.92}$ | $88.00_{\uparrow1.33}$ | $81.50_{\uparrow2.58}$ | $89.08_{\uparrow2.00}$ | 87.26 |
| AFlow | ✓ | $81.80_{\uparrow3.26}$ | $91.16_{\uparrow3.71}$ | $96.22_{\downarrow0.63}$ | $88.33_{\uparrow1.66}$ | $80.90_{\uparrow1.98}$ | $90.93_{\uparrow3.85}$ | 88.22 |
| G-Designer | ✓ | $80.39_{\uparrow1.85}$ | $91.09_{\uparrow3.64}$ | $97.78_{\uparrow0.93}$ | $90.00_{\uparrow3.33}$ | $80.75_{\uparrow1.83}$ | $89.37_{\uparrow2.29}$ | 88.23 |
| AgentPrune | ✓ | $82.40_{\uparrow3.86}$ | $91.92_{\uparrow4.47}$ | $97.88_{\uparrow1.03}$ | $90.37_{\uparrow3.70}$ | $80.93_{\uparrow2.01}$ | $87.17_{\uparrow0.09}$ | 88.22 |
| GTD | ✓ | $79.41_{\uparrow0.87}$ | $91.38_{\uparrow3.93}$ | $96.20_{\downarrow0.65}$ | $90.24_{\uparrow3.57}$ | $79.44_{\uparrow0.52}$ | $87.86_{\uparrow0.78}$ | 87.42 |
| MaAS | ✓ | $82.32_{\uparrow3.78}$ | $91.13_{\uparrow3.68}$ | $98.08_{\uparrow1.23}$ | $89.65_{\uparrow2.98}$ | $80.25_{\uparrow1.33}$ | $89.57_{\uparrow2.49}$ | 88.50 |
| ARG-Designer | ✓ | $79.10_{\uparrow0.56}$ | $91.25_{\uparrow3.80}$ | $98.55_{\uparrow1.70}$ | $92.21_{\uparrow5.54}$ | $81.10_{\uparrow2.18}$ | $89.19_{\uparrow2.11}$ | 88.57 |
| RADAR | ✓ | $\mathbf{83.66}_{\uparrow5.12}$ | $\mathbf{92.51}_{\uparrow5.06}$ | $\mathbf{98.81}_{\uparrow1.96}$ | $\mathbf{93.26}_{\uparrow6.59}$ | $\mathbf{82.84}_{\uparrow3.92}$ | $\mathbf{91.28}_{\uparrow4.20}$ | **90.32** |

ing their roles to a liar agent that intentionally provides false information. As demonstrated in Figure 4, while the degree of degradation varies across methods, most frameworks exhibit substantial performance drops under such attacks. For example, the complete graph topology suffers significant degradation, with performance decreasing by up to 4.47%, while ARG-Designer demonstrates relatively strong robustness, with only a modest performance drop of approximately 1.05%. In contrast, RADAR exhibits exceptional resilience to adversarial perturbations, maintaining nearly identical performance before and after the attack.

## 5.7. Ablation Studies

**Impact of the key components.** We evaluate three variants of our proposed RADAR across three datasets. Specifically, (1) *w/o ES* removes the effective size component from both the ordering network and the denoising network; (2) *w/o utility* eliminates the utility loss defined in Equation (6); (3) *w/o query* removes the task query input from the denoising network; (4) *ON w/o ES* and *DN w/o ES* removes the effective size component from the ordering and denoising network respectively and (5) *non-diffusion* denotes a baseline that employs non-diffusion techniques (e.g., ARG-Designer). As shown in Table 2, removing any component consistently degrades the model's performance. In particular, excluding the task query impairs the model's task adaptiveness, while removing the effective size component results in a

*Table 2.* Ablation studies on three datasets. "ON" denotes ordering network and "DN" indicates denoising network.

| Variants | MMLU | GSM8K | MultiArith |
|---|---|---|---|
| RADAR | 83.66 | 92.51 | 98.81 |
| *w/o* ES | $81.05_{\downarrow2.61}$ | $91.22_{\downarrow1.29}$ | $98.31_{\downarrow0.50}$ |
| *w/o* utility | $82.96_{\downarrow0.70}$ | $92.02_{\downarrow0.49}$ | $98.47_{\downarrow0.34}$ |
| *w/o* query | $79.08_{\downarrow4.58}$ | $91.82_{\downarrow0.69}$ | $97.81_{\downarrow1.00}$ |
| ON *w/o* ES | $79.74_{\downarrow3.92}$ | $91.96_{\downarrow0.55}$ | $98.47_{\downarrow0.34}$ |
| DN *w/o* ES | $80.39_{\downarrow3.27}$ | $92.37_{\downarrow0.14}$ | $98.01_{\downarrow0.80}$ |
| non-diffusion | $79.10_{\downarrow4.56}$ | $91.25_{\downarrow1.26}$ | $98.55_{\downarrow0.26}$ |

systematic decline in performance. Similarly, replacing the diffusion-based generation mechanism also leads to noticeable degradation, highlighting its importance in modeling structural dependencies. These results empirically validate the effectiveness of our design.

**Impact of the hyper-parameters.** We analyze the influence of two factors: the number of agents and the choice of LLMs. As presented in Figure (5a), when the number of agents increases, performance initially improves; however, the marginal gains diminish beyond a certain point. For complex tasks like MultiArith, increasing the number of agents can lead to continued performance improvements. In contrast, for simpler tasks like MMLU, adding more agents may introduce unnecessary redundancy and even degrade performance. Figure (5b) further examines the transferabil-

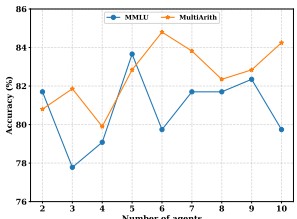

*(a)* Impact of agent numbers.

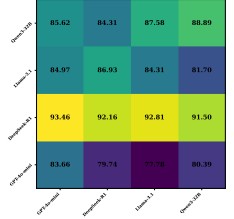

*(b)* LLM models on MMLU.

*Figure 5.* Different number of agents and LLM models.

*Table 3.* Multi-agent collaboration vs. single-agent performance.

| LLMs | Single Agents | RADAR(MAS) |
|---|---|---|
| gpt-4o-min | 78.54 | 83.66$_{\uparrow 5.12}$ |
| Llama-3.1-70B | 77.12 | 84.31$_{\uparrow 7.19}$ |
| Qwen3-32B | 84.97 | 88.89$_{\uparrow 3.92}$ |
| DeepSeek-R1 | 90.81 | 92.16$_{\uparrow 1.35}$ |

ity of the generative model across different LLM backbones on the MMLU (i.e., `gpt-4o-mini`, `Llama-3.1-70B Instruct`, `Qwen3-32B` and `DeepSeek-R1`). The `x-axis` shows the training LLM, and the `y-axis` shows the evaluation LLM. The results show that DeepSeek-R1 exhibits strong reasoning capabilities, even when the generative model is trained with a comparatively weaker LLM. Thus, our model enables training with weaker, more cost-effective LLMs while leveraging stronger, more expensive models during evaluation and deployment.

**Impact of multi-agent collaboration framework.** To verify the effectiveness of multi-agent collaboration over single-agent settings, we evaluate RADAR across a range of base models with varying capabilities, including stronger models such as DeepSeek-R1. The results are summarized in Table 3. We observe that RADAR consistently improves performance across all models, including stronger ones. While the absolute gains become smaller for more capable models (e.g., +1.35 for DeepSeek-R1 vs. larger gains for weaker models), the improvements remain stable. This suggests that RADAR provides complementary benefits beyond base model capability, such as structured collaboration and reduced redundancy, which remain useful even when single-agent performance is strong.

**Efficiency.** We provide a comparison of training time, inference time, and overall token consumption on GSM8K dataset below. As shown in Table 4, RADAR reduces token consumption, while incurring a moderate increase in inference time due to the additional graph generation process. Training time remains comparable to existing methods. Regarding inference time, AFlow is a search-based method that learns a single optimized workflow shared across all queries, making it faster at inference. In contrast, our method generates query-adaptive topologies for each instance. While

*Table 4.* Comparison of wall-clock time and token consumption.

| Methods | Overall Token | Train | Infer |
|---|---|---|---|
| AFlow | $1.4 \times 10^7$ | 2h43min | 7.32min |
| AgentPrune | $1.1 \times 10^7$ | 1h27min | 14.25min |
| RADAR | $6.5 \times 10^6$ | 2h10min | 17.55min |

*Table 5.* Distribution of graph sizes (GS), densities, and effective sizes (ES). ARG is short for ARG-Designer.

| Methods | GS | Densities | ES |
|---|---|---|---|
| GDesigner | 4.92±0.25 | 0.302±0.065 | 0.73±0.17 |
| ARG | 3.79±1.57 | 0.317±0.133 | 0.68±0.13 |
| RADAR | 4.60±0.66 | 0.289±0.071 | 0.92±0.18 |

this per-query adaptation introduces additional overhead, it enables more efficient communication tailored to task complexity.

**Distribution of Generated Collaboration Graphs Across Models.** We analyze the distribution of generated graph structures by reporting the mean and standard deviation of graph size, density, and effective size across methods on MMLU dataset. We note that although the number of agents is set to 5, some nodes may remain inactive due to isolation; we therefore report statistics over the active subgraph during execution. From these results in Table 5, RADAR exhibits two key structural differences: (1) Higher effective size (lower redundancy). RADAR achieves significantly higher effective size (0.92 vs. 0.73 / 0.68), indicating that it produces less redundant and more informative communication structures. (2) More efficient connectivity. RADAR maintains comparable graph size and slightly lower density, suggesting that it avoids unnecessary connections while preserving sufficient coordination. In contrast, baseline methods tend to produce either denser or less structured topologies with lower effective size, leading to more redundant communication.

## 6. Conclusion

In this paper, we propose an iterative framework for multi-agent collaboration topology design, which provides high accuracy, low token consumption, and strong robustness across diverse scenarios. Specifically, we employ a graph diffusion model that leverages graph effectiveness signals to iteratively construct collaboration topologies, explicitly modeling redundancy as part of the topology generation process. The proposed framework is flexible and task-adaptive, enabling effective topology customization for different queries. We conduct extensive experiments on six benchmarks spanning general reasoning, code generation and mathematical problem solving, which collectively demonstrate the effectiveness of our method. Extending the framework to support dynamic role composition or generation is an interesting direction for future work.

## Acknowledgements

This work is sponsored by CCF-Tencent Rhino-Bird Open Research Fund and is supported by the "111 Center" (No. B26023).

## Impact Statement

**Ethical Aspects.** We do not identify any ethical concerns arising from the motivation, methodology, experimental setup, or data usage in this work. The proposed RADAR framework is designed to advance research in multi-agent systems and automated communication topology design in a responsible manner, with the goal of improving efficiency and robustness in collaborative AI systems.

**Societal Consequences.** In this work, we introduce a new paradigm for multi-agent system design based on iterative, redundancy-aware topology generation. By enabling fine-grained and task-adaptive resource allocation, RADAR improves efficiency while maintaining high output quality. Reduced inference costs and increased flexibility in multi-agent workflows may help lower barriers to adopting intelligent automation, with potential benefits for a broad range of applications in education, scientific research, and industry.

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

*Table 6.* Dataset statistics.

| Category | Dataset | Format | Metric | #Samples |
|----------|---------|--------|--------|----------|
| General Reasoning | MMLU | Multi-choice | Accuracy | 153 |
| Math Solving | GSM8K | Number | Accuracy | 1,319 |
| | MultiArith | Number | Accuracy | 600 |
| | SVAMP | Number | Accuracy | 1,000 |
| | AQuA | Multi-choice | Accuracy | 254 |
| Code Generation | HumanEval | Code | Pass@1 | 164 |

## A. Dataset Statistics

We present the statistics of three types of datasets in Table 6.

## B. Baselines

The configurations used for each baseline method are described in below:

- **CoT (Wei et al., 2022).** Chain-of-Thought (CoT) prompting enables LLM agents to perform step-by-step reasoning instead of generating direct answers. We follow the implementation described in (Zhang et al., 2023).

- **ComplexCoT (Fu et al., 2023).** Our experiments are based on the official implementation released at `https://github.com/FranxYao/Complexity-Based-Prompting/tree/main`.

- **SC(COT×5) (Wang et al., 2023).** For robustness, we aggregate five solutions produced using Chain-of-Thought prompting.

- **MultiPersona (Wang et al., 2024b).** It converts a single LLM into a cognitive synergist through multi-turn self-collaboration with multiple personas. We utilize the official code available at `https://github.com/MikeWangWZHL/Solo-Performance-Prompting`.

- **LLM-Debate (Du et al., 2023).** We employ five role-specialized LLM agents that participate in up to two debate rounds, with the final output selected by majority voting. The implementation is based on `https://github.com/ucl-dark/llm_debate`.

- **LLM-Blender (Jiang et al., 2023).** In experiments, LLM-Blender is instantiated using two `gpt-4o-mini` models, one `Qwen-2.5-72B`, and one `LLaMA-3.1-70B`. The source code is available at `https://github.com/yuchenlin/LLM-Blender`.

- **DyLAN (Liu et al., 2024c).** Our experiments are conducted using the implementation available at `https://github.com/SALT-NLP/DyLAN`.

- **AgentVerse (Chen et al., 2024c).** We follow the original implementation from `https://github.com/OpenBMB/AgentVerse`.

- **MacNet (Qian et al., 2025).** We use the "MacNet-MESH" variant of MacNet, which employs a fully connected network topology. The implementation is available at `https://github.com/OpenBMB/ChatDev/tree/macnet`.

- **AutoAgents (Chen et al., 2024a).** We adopt the configuration detailed in `https://github.com/Link-AGI/AutoAgents`.

- **GPTSwarm (Zhuge et al., 2024).** The method is executed using the original configuration as specified in `https://github.com/metauto-ai/GPTSwarm`.

- **ADAS (Hu et al., 2025).** Our implementation is based on the authors' released code in `https://github.com/ShengranHu/ADAS`.

*Table 7.* GNN backbone for different components.

| Variants | MMLU | MultiArith |
|---|---|---|
| Ordering (RGAT), Denoising (GAT) | 82.35 | 97.97 |
| Ordering (RGAT), Denoising (GCN) | 82.30 | 98.14 |
| Ordering (RGCN), Denoising (GCN) | 82.23 | 98.64 |
| Ordering (RGCN), Denoising (GAT) | 83.66 | 98.81 |

- **AgentSquare (Shang et al., 2025).** We build on the modular search framework from (Shang et al., 2025), using `GPT-4o-mini` as the fixed base LLM. Early stopping is applied, with training halted after five iterations without improvement. The codes are available at `https://github.com/tsinghua-fib-lab/AgentSquare`.

- **AFlow (Zhang et al., 2025d).** To preserve experimental fairness under identical conditions, we configure AFlow to use `GPT-4o-mini` and limit the number of iterations to 20. We use the source code released at `https://github.com/FoundationAgents/AFlow`.

- **G-Designer (Zhang et al., 2025c).** It employs a variational graph auto-encoder to construct the communication topology. We utilize the source code released at `https://github.com/yanweiyue/GDesigner`.

- **AgentPrune (Zhang et al., 2025b).** It performs pruning on the spatiotemporal message-passing graph. We utilize the source code available at `https://github.com/yanweiyue/AgentPrune`.

- **GTD (Jiang et al., 2025).** We implement the method according to the original configuration specified in the paper.

- **MaAS (Zhang et al., 2025a).** It samples query-dependent agentic systems from the supernet. We use the source code available at `https://github.com/bingreeky/MaAS/`.

- **ARG-Designer (Li et al., 2025).** It dynamically determines the number of agents, assigns roles from an extensible pool, and establishes optimal communication links. We use the implementation available at `https://github.com/Shiy-Li/ARG-Designer`.

## C. More Ablation Studies

We present additional ablation studies in Figure 6. Regarding the trade-off parameter $\beta$, which balances outgoing and incoming effective size, we assign greater weight to outgoing effective size. Outgoing redundancy affects multiple downstream agents simultaneously and thus dominates system-level cost, whereas incoming redundancy primarily impacts a single agent. Based on this analysis, we consistently set $\beta$ to either $0.7$ or $0.8$ to achieve a favorable performance. We also analyze the impact of the number of layers in the denoising network. The performance generally improves as the number of layers increases, but begins to decline beyond a certain point, indicating diminishing returns from overly complex architectures. Since generating excessively complicated structures is unnecessary, we set the number of layers to 3 or 4 for most datasets.

We also explored alternative GNN architectures for both ordering network and denoising network. Specifically, we replaced the RGCN in the ordering network with RGAT, and the GAT in the denoising network with GCN. The results in Table 7 indicate that these alternatives achieve comparable but generally slightly worse performance than our default design. This suggests that while the overall framework is not highly sensitive to the specific GNN choice, the selected combination (RGCN for ordering and GAT for denoising) provides a better balance between relational modeling and attention-based refinement.

Meanwhile, we extended our robustness evaluation to include both structural attacks and combined prompt + structural attacks, in addition to the original prompt-based setting. For prompt attacks, we simulate system prompt perturbations by converting two of the five agents into liar agents. For structural attacks, we inject noise into the collaboration graph by randomly adding 50% additional edges. As shown in the Table 8, our method maintains consistent performance across prompt, structural, and combined attacks, indicating robustness to multiple sources of perturbation.

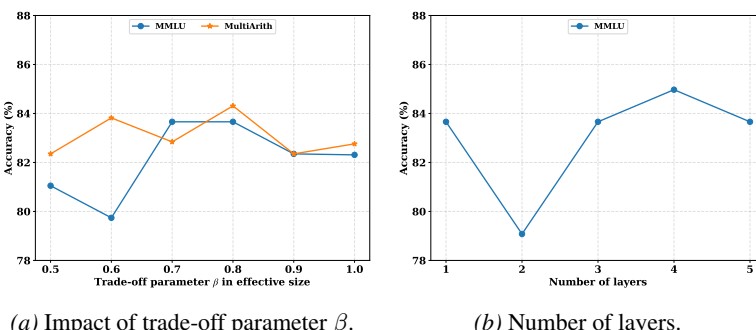

*(a)* Impact of trade-off parameter $\beta$.    *(b)* Number of layers.

*Figure 6.* Hyper-parameter sensitivity analysis.

*Table 8.* Model performance under different attacks.

| Variants | MMLU | GSM8K | MultiArith |
|---|---|---|---|
| Prompt Attack | 83.14 | 92.12 | 98.13 |
| Structure Attack | 82.08 | 91.82 | 98.64 |
| Prompt & Structure Attack | 81.88 | 91.46 | 97.79 |
| RADAR | 83.66 | 92.51 | 98.81 |

## D. Case Study

Figure 7 illustrates the communication topologies generated by RADAR. As we can see, the resulting structures are task-adaptive and vary across different tasks and datasets. Although the maximum number of agents is fixed at five, RADAR does not necessarily activate all agents for simpler tasks. In contrast, for more complex benchmarks such as AQUA and HumanEval, all five agents are utilized in the generated topology, reflecting the increased reasoning and coordination requirements of these tasks.

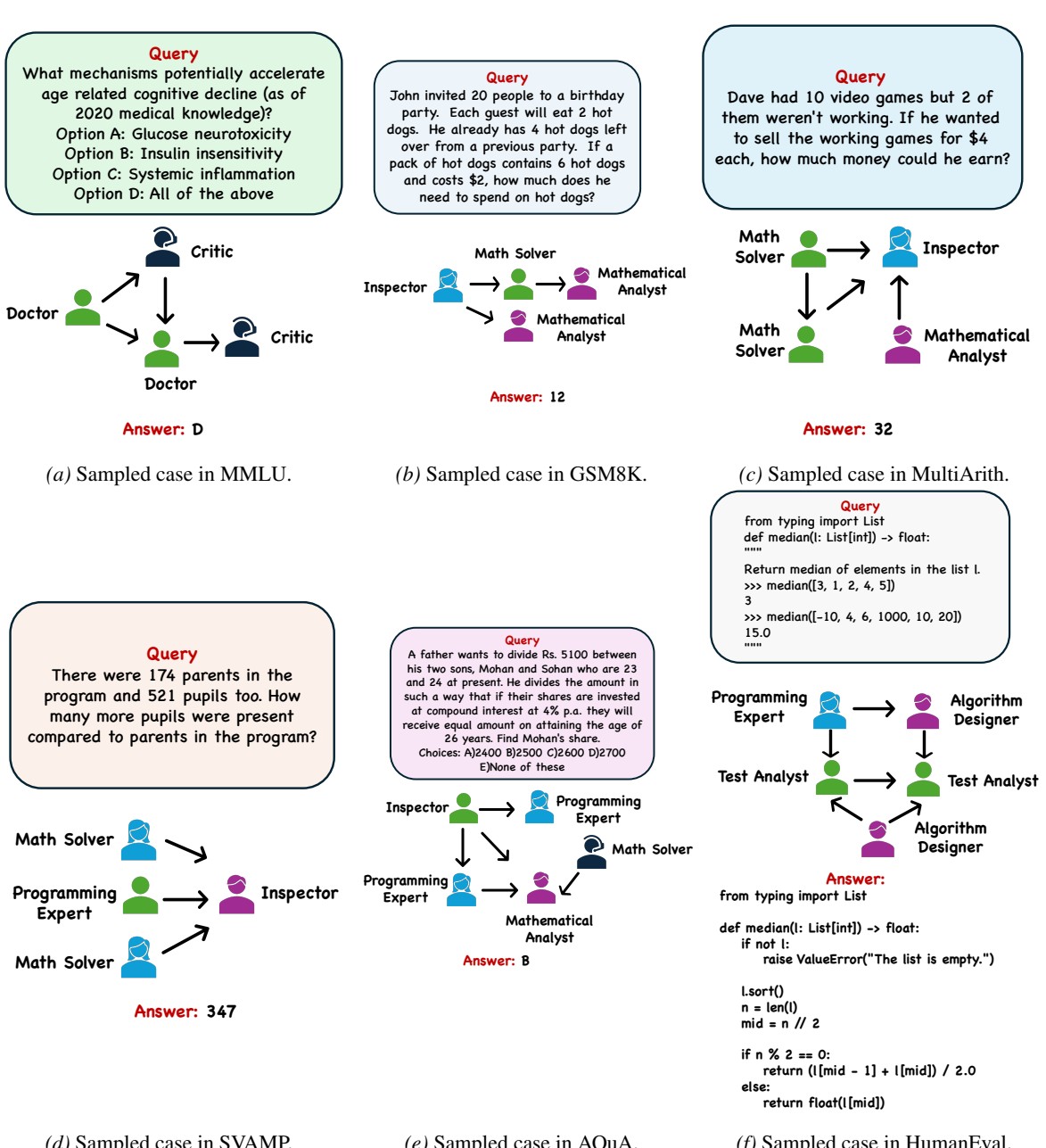

*(a)* Sampled case in MMLU.

*(b)* Sampled case in GSM8K.

*(c)* Sampled case in MultiArith.

*(d)* Sampled case in SVAMP.

*(e)* Sampled case in AQuA.

*(f)* Sampled case in HumanEval.

*Figure 7.* Case study of the communication topologies generated by RADAR.

