# OpenReview forum: "RADAR: Redundancy-Aware Diffusion for Multi-Agent Communication Structure Generation"
_ICML.cc/2026/Conference — ICML 2026 regular_

### Official Review · Reviewer_gtQ6 · 2026-02-25

**Soundness:** 3
**Presentation:** 2
**Significance:** 2
**Originality:** 3
**Overall Recommendation:** 4
**Confidence:** 3

**Summary:**

This paper proposes RADAR, a framework for automatically generating multi-agent communication topologies using conditional discrete graph diffusion models. The key idea is to incorporate the concept of "effective size" from social network theory into both the forward diffusion ordering and the reverse denoising process, so that the generated collaboration graphs exhibit low redundancy. Specifically, a redundancy-aware ordering network prioritizes node masking based on effective size during the forward process, while a GNN-based denoising network reconstructs the graph conditioned on the task query during the reverse process. The model is trained jointly via variational lower bound optimization and REINFORCE-based policy gradients, with periodic task utility feedback. Experiments on six benchmarks show that RADAR outperforms a range of baselines.

**Compliance With Llm Reviewing Policy:**

Affirmed.

**Final Justification:**

The rebuttal addresses most of my concerns.

However, the paper still has many areas that need improvement. Given its current level of maturity, I will raise my score to "weak accept."

**Key Questions For Authors:**

1. The reverse process predicts both agent roles and edge connectivity (Section 4.2). Is the role space fixed and predefined per dataset, or can RADAR compose novel roles?

2. Regarding the effective size formulation in Equation (7): the indicator function $\mathbb{I}[r(j) \neq r(q)]$ penalizes connections between agents of *different* roles, yet intuitively, agents sharing the *same* role would contribute more redundant information. Could the authors clarify the rationale behind this design?

3. All main experiments use gpt-4o-mini, while Figure 5b only evaluates transferability on MMLU. Do the authors expect RADAR's advantages to hold with stronger base models where single-agent performance is already high, and the marginal benefit of multi-agent collaboration may diminish?

**Limitations:**

yes

**Strengths And Weaknesses:**

## Strengths:
1. Framing multi-agent topology design as an iterative graph diffusion problem is a natural choice. The integration of effective size as a redundancy-aware inductive bias into both the ordering and denoising networks is theoretically grounded and interesting.
2. RADAR achieves the best accuracy across all six benchmarks while consuming significantly fewer tokens, acting effectively and efficiently.

## Weaknesses:

1. Since RADAR generates a new communication topology for each incoming query via iterative diffusion (GNN forward passes, multiple sampling trajectories), the latency overhead could be non-trivial. The paper only reports token consumption but never quantifies the time cost of the topology generation process itself. For practical deployment, knowing the overhead is critical.

2. Beyond the six-case qualitative study in Figure 7, the paper provides no systematic characterization of what RADAR actually learns. Key questions remain unanswered: What is the distribution of graph sizes, densities, and effective sizes across datasets? How do RADAR's generated topologies structurally differ from those produced by G-Designer, ARG-Designer, or GTD? Without such analysis, it is difficult to understand *why* RADAR outperforms existing methods.

3. **(Minor, more like a question.)** Effective size is a purely structural metric, independent of query semantics. The effective size (Eq. 7–8) measures topological redundancy based solely on graph connectivity and role labels, without considering the task query. This means the redundancy-aware ordering in the forward process is query-agnostic, even though different queries may tolerate or benefit from different levels of redundancy.

4. The GNN-based ordering and denoising networks are trained on graphs constructed from a fixed set of agents, roles, and queries per dataset. It is unclear how RADAR would perform when encountering queries substantially different from the training distribution, or when new agent roles not seen during training are introduced. GNNs are known to struggle with out-of-distribution generalization, and the paper provides no evidence of robustness to distributional shift beyond the adversarial prompt attack experiment.

5. No variance or confidence intervals are reported in Table 1. Given that performance differences among top methods are often within 1–2%, and considering stochasticity in both LLM outputs and diffusion sampling, it is difficult to assess whether improvements are statistically meaningful.

6. **(Minor)** Typos: ""liner exchange" (line 25) -> linear,  "the extend to which" (line 185). -> extent.
The paper would benefit from careful proofreading; there are more typos, actually.

---

> ### Author Rebuttal · Authors · 2026-03-30
>
> *Q1. The time cost of the topology generation process itself.*
>
> A1. We provide a comparison of training time, inference time, and overall token consumption on GSM8K dataset below.
>
> |Variants|Overall Token|Train Time|Infer Time|
> |:-|:-:|:-:|:-:|
> |AFlow|$1.4 \times10^7$|2h 43 min|7.32 min|
> |AgentPrune|$1.1 \times 10^7$|1h 27 min|14.25 min|
> |RADAR|$6.5 \times 10^6$|2h 10 min|17.55 min|
>
> RADAR reduces token usage with similar training time but slightly higher inference cost due to per-query graph generation. Unlike AFlow’s fixed workflow, RADAR adapts topology per query, enabling more efficient communication. We will include this discussion in the final version.
> ***
> *Q2. What is the distribution of graph sizes, densities, and effective sizes across datasets?*
>
> A2. We analyze the distribution of generated graph structures by reporting the mean and standard deviation of graph size, density, and effective size across methods on MMLU dataset:
>
> |Variants|graph sizes|densities|effective sizes|
> |:-|:-:|:-:|:-:|
> |GDesigner|4.92$\pm$0.25|0.302$\pm$0.065|0.73$\pm$0.17|
> |ARG-Designer|3.79$\pm$1.57|0.317$\pm$0.133|0.68$\pm$0.13|
> |RADAR|4.60$\pm$0.66|0.289$\pm$0.071|0.92$\pm$0.18|
>
> Although up to 5 agents are allowed, some nodes may remain inactive due to isolation; we therefore report statistics over the active subgraph during execution. RADAR exhibits higher effective size (0.92 vs. 0.73 / 0.68), indicating lower redundancy, while maintaining comparable size and slightly lower density, reflecting more efficient connectivity. We will include this analysis in the final version.
> ***
> *Q3. (Minor, more like a question.) Effective size is a purely structural metric.*
>
> A3. In current framework, effective size is a query-agnostic structural metric, while query dependence is introduced through conditional generation. It serves as a general regularizer to reduce redundancy, since different queries may benefit from different redundancy levels. Incorporating query-aware redundancy is a promising direction for future work, which we will explore in the future work.
> ***
> *Q4. The paper provides no evidence of robustness to distributional shift beyond the adversarial prompt attack experiment.*
>
> A4. We agree that distributional generalization is important. While the role space is fixed and unseen roles are not supported, our method shows robustness to new queries through query-conditioned generation and structural priors (e.g., effective size). Empirically, it maintains stable performance across datasets and attack settings (**see A5 in Reviewer m9oe**), though this does not fully cover all forms of distribution shift. Handling stronger shifts is left for future work.
> ***
> *Q5. No variance or confidence intervals are reported in Table 1.*
>
> A5. We observe that the variance across runs is relatively small, and the ranking between methods remains consistent. For example, RADAR achieves 92.51$\pm$0.36 compared to 91.25$\pm$0.74 for the ARG-Designer baseline on GSM8K dataset, where the performance gap exceeds the observed variance. We will include these statistics (mean $\pm$ std) in Table 1 in the final version.
> ***
> *Q6. Is the role space fixed and predefined per dataset, or can RADAR compose novel roles?*
>
> A6. In the current formulation, the role space is fixed and predefined, and RADAR does not compose novel roles. Specifically, the model predicts role assignments from a predefined set, while dynamically determining how these roles are organized and connected during the generation process. Extending the framework to support dynamic role composition or generation is an interesting direction for future work.
> ***
> *Q7. Could the authors clarify the rationale behind Equation (7)?*
>
> A7. Thanks for pointing this out. *This is a typo in the paper*. The correct formulation considers two in-neighbor nodes to be redundant when they are adjacent and have the **same roles**, not different roles. We confirm that our implementation and all experiments follow the correct definition (same-role redundancy), and the reported results are unaffected. This is consistent with our released anonymous code (see denoising.py and ordering.py in https://anonymous.4open.science/r/RADAR-8430). We will fix this typo and clarify the definition in the final version.
> ***
> *Q8. With stronger base models, will the benefit of multi-agent collaboration diminish?*
>
> A8. We evaluate RADAR across a range of base models with varying capabilities, including stronger models such as DeepSeek-R1. The results are summarized below:
>
> |LLMs|gpt-4o-mini|llama-3.1-70b|qwen3-32b|deepseek-r1|
> |:-|:-:|:-:|:-:|:-:|
> |Single Agent|78.54|77.12|84.97|90.81|
> |Multi-Agents (RADAR)|83.66|84.31|88.89|92.16|
>
> RADAR consistently improves performance across all models, including stronger ones. While gains diminish with stronger models (e.g., +1.35 for DeepSeek-R1 vs. larger gains for weaker models), they remain stable, indicating complementary benefits from structured collaboration and reduced redundancy.

---

> > ### Author Rebuttal · Reviewer_gtQ6 · 2026-04-02
> >
> > Most of my problems have been resolved.
> > Hope the modifications could be made to the final version, especially for the typos.

---

> > > ### Author Response · Authors · 2026-04-02
> > >
> > > Thank you for raising the score. We will include these details and fix the typos in the final version.

---

### Official Review · Reviewer_kigm · 2026-03-06

**Soundness:** 3
**Presentation:** 3
**Significance:** 3
**Originality:** 3
**Overall Recommendation:** 5
**Confidence:** 3

**Summary:**

In this paper, the authors propose RADAR, a new generative framework for automating multi-agent communication topologies in LLM-based systems. To overcome the high token costs and redundant data flows associated with fixed or single-step structures, the approach utilises conditional discrete graph diffusion models to build these networks step-by-step. Through rigorous evaluation across six diverse benchmarks, RADAR establishes state-of-the-art accuracy, better token efficiency, and greater defence against adversarial agent attacks when evaluated against traditional heuristic, search-based, and learning-based multi-agent models.

**Compliance With Llm Reviewing Policy:**

Affirmed.

**Final Justification:**

I thank the authors for their detailed and comprehensive rebuttal. My concerns have been thoroughly resolved point by point. Therefore, I am raising my score to 5. Please ensure that all the details provided in this rebuttal are incorporated into the final camera-ready version of the paper as promised.

**Key Questions For Authors:**

I would be happy to raise my score if the authors can address the following questions:

1. In Section 4, the paper states that "multiple baseline topologies are constructed to form a foundational dataset" to train the graph diffusion model. Could you elaborate on the exact mechanism, scale, and computational cost of generating this offline dataset? How sensitive is the final RADAR model to the quality of this initial dataset?

2. How does the wall-clock training and inference overhead of RADAR compare to baselines like AgentPrune or AFlow? A brief discussion on the computational trade-off between generating the graph vs. the token savings during execution would be beneficial.

3. In Section 5.3, the pipeline specifies using a 3-layer RGCN for the ordering network and a 3-layer GAT for the denoising network. Have the authors experimented with other GNN architectures? Recent work, such as [1] demonstrates that different GNN architectures exhibit varying capabilities when modelling LLM multi-agent systems. An ablation or discussion on how the specific choice of GNN impacts the generated topologies and downstream token efficiency would strengthen the methodology.

[1] Zhang, Yuanshuo, et al. "FLORA: GNNs as Predictors of Agentic Workflow Performances." The Fourth Learning on Graphs Conference.

**Limitations:**

yes

**Strengths And Weaknesses:**

Strengths:

1. The adaptation of Burt's "effective size" to quantify incoming and outgoing redundancy in a directed multi-agent graph is intuitive and mathematically well-grounded. The training regime, which employs the REINFORCE algorithm to optimise the non-differentiable discrete ordering network alongside a policy gradient estimator for task utility, is robust and appropriately chosen for this architecture.

2. As LLM multi-agent systems scale, communication bottlenecks and token costs are becoming critical deployment barriers. RADAR addresses a highly relevant problem.

3. The paper is well-structured and clearly written.

Weaknesses:

1. The initial data collection process is somewhat opaque. Section 4 briefly mentions that "multiple baseline topologies are constructed to form a foundational dataset," but the paper lacks specifics on the computational cost and scale of this bootstrapping phase.

2. The transition between the pre-training on the "foundational dataset" and the RL-based fine-tuning using task utility could be articulated more explicitly in Section 4.3.

---

> ### Author Rebuttal · Authors · 2026-03-30
>
> We sincerely thank the Reviewer kigm for the thorough and insightful feedback. Our responses are as follows:
> ***
> *Q1. Could you elaborate on the exact mechanism, scale, and computational cost of generating this offline dataset? How sensitive is the final RADAR model to the quality of this initial dataset?*
>
> A1. We thank the reviewer for this question. We clarify the mechanism, scale, computational cost, and sensitivity of the offline dataset generation below.
>
> **Mechanism**. The initial dataset is constructed by instantiating a diverse set of baseline communication topologies (e.g., fully connected, mesh, star, layered, and random) with varying numbers of agents (e.g., 3 or 4 agents). For each configuration, we randomly assign agent roles and evaluate the resulting multi-agent system on sampled training instances (e.g., whether the final answer is correct or incorrect). This provides a diverse set of graph-structured samples to initialize the diffusion model.
>
> **Scale**. In our implementation, we consider multiple topology families and small agent counts (e.g., 3–4 agents), resulting in a modest number of configurations (e.g., 10 in total). For each configuration, we sample a subset of training instances (controlled by a train_set_size parameter, i.e., 50), making the overall dataset size manageable (i.e., 500 training samples).
>
> **Computational Cost**. The offline dataset generation is a one-time preprocessing step. Its cost scales linearly with the number of configurations and sampled training instances. In practice, this step is comparable to standard multi-agent evaluation runs and does not introduce additional overhead during training or inference.
>
> **Sensitivity**. We find that the final RADAR model is not highly sensitive to the exact choice of initial configurations. *In implementation, we use the same configurations across all six datasets (e.g., fully connected, mesh, star, layered, and random), since they cover a diverse range of topology patterns*. The diffusion-based generation further refines these initial structures during training, reducing dependence on any specific initialization. We will clarify this process and its role in the final version.
> ***
> *Q2. How does the wall-clock training and inference overhead of RADAR compare to baselines like AgentPrune or AFlow? A brief discussion on the computational trade-off between generating the graph vs. the token savings during execution would be beneficial.*
>
> A2. We thank the reviewer for this important question. We provide a comparison of training time, inference time, and overall token consumption on GSM8K dataset below.
>
> | Variants | Overall Token | Train Time | Infer Time |
> |:---:|:---:|:---:|:---:|
> | AFlow | $1.4 \times 10^7$ | 2h 43 min | 7.32 min |
> | AgentPrune | $1.1 \times 10^7$ | 1h 27 min | 14.25 min |
> | RADAR | $6.5 \times 10^6$ | 2h 10 min | 17.55 min |
>
> As shown in the table, RADAR reduces token consumption, while incurring a moderate increase in inference time due to the additional graph generation process. Training time remains comparable to existing methods. Regarding inference time, AFlow is a search-based method that learns a single optimized workflow shared across all queries, making it faster at inference. In contrast, our method generates query-adaptive topologies for each instance. While this per-query adaptation introduces additional overhead, it enables more efficient communication tailored to task complexity. We will include this discussion in the final version.
> ***
> *Q3. In Section 5.3, the pipeline specifies using a 3-layer RGCN for the ordering network and a 3-layer GAT for the denoising network. Have the authors experimented with other GNN architectures?*
>
> A3. We thank the reviewer for this question. We have explored alternative GNN architectures for both components. Specifically, we replaced the RGCN in the ordering network with RGAT, and the GAT in the denoising network with GCN.
>
> | Variants | MMLU | MultiArith |
> |:---:|:---:|:---:|
> | Ordering (RGAT), Denoising (GAT) | 82.35 | 97.97 |
> | Ordering (RGAT), Denoising (GCN) | 82.30 | 98.14 |
> | Ordering (RGCN), Denoising (GCN) | 82.23 | 98.64 |
> | Ordering (RGCN), Denoising (GAT) | 83.66 | 98.81 |
>
> The results (shown above) indicate that these alternatives achieve comparable but generally slightly worse performance than our default design. This suggests that while the overall framework is not highly sensitive to the specific GNN choice, the selected combination (RGCN for ordering and GAT for denoising) provides a better balance between relational modeling and attention-based refinement. We will include these comparisons in the final version for completeness.
> ***

---

> > ### Author Rebuttal · Reviewer_kigm · 2026-04-02
> >
> > I thank the authors for their detailed and comprehensive rebuttal. My concerns have been thoroughly resolved point by point. Therefore, I am raising my score to 5. Please ensure that all the details provided in this rebuttal are incorporated into the final camera-ready version of the paper as promised.

---

> > > ### Author Response · Authors · 2026-04-02
> > >
> > > Thank you for raising the score. We will certainly incorporate these details into the final version.

---

### Official Review · Reviewer_Pm9S · 2026-03-11

**Soundness:** 3
**Presentation:** 3
**Significance:** 3
**Originality:** 3
**Overall Recommendation:** 5
**Confidence:** 4

**Summary:**

This paper proposes a discrete graph diffusion method to train an automatic design model for Multi-Agent Systems. The authors introduce "effective size" as a metric to quantify the degree of redundancy in the MAS topology. Based on this metric, a redundancy-aware ordering network is utilized to adjust priority weights during the graph diffusion process. The trained MAS automatic design model achieves outstanding task performance while consuming fewer tokens compared to baseline methods.

**Compliance With Llm Reviewing Policy:**

Affirmed.

**Key Questions For Authors:**

1. Regarding Equation 7 , if I understand correctly, a larger effective size indicates lower redundancy. However, there is a counter-intuitive aspect to this formulation: why is the structure considered non-redundant only when two in-neighbor nodes are adjacent and have different roles ($A_{jq}=1$ and $r(j)\neq r(q)$)? Intuitively, shouldn't they be considered non-redundant when they are not adjacent and have different roles?
2. In Equation 10 , how is the embedding vector $h_t$ added to the scalar $\phi(v_t)$ and directly involved in the softmax operation? Is there a missing projection step here to align the dimensions?
3. In the text below Equation 12, should the bias term $\psi(v_i)$ in the update rule $h_i^L=h_i^L+\psi(v_i)$ actually be the effective size $\phi(v_i)$? Similar to Q2, how is the addition between a vector and a scalar computed in this specific context?
4. The authors claim that existing methods like AgentPrune and AgentDropout optimize MAS efficiency in an inefficient "post hoc" manner because their agent roles are fixed. How are the agent roles determined in the proposed method? Does it involve the dynamic optimization of their profiles or prompts?

**Limitations:**

1. As shown in Figure 5(a), the optimal number of agents varies depending on the difficulty of the task. The current paper treats this number as a manually determined hyperparameter, which is time-consuming and lacks reliability. The paper could be improved by extending the method to dynamically determine the number of agents. For example, introducing an early-stopping mechanism during the reverse diffusion process could allow the model to judge whether the current MAS structure is sufficient for the given task, thereby adaptively adjusting the scale of the MAS and enhancing its overall scalability.

**Strengths And Weaknesses:**

### Strengths
1. The paper cleverly formulates the automatic design of MAS using a discrete graph diffusion mathematical model and innovatively incorporates "effective size" as a quantitative metric for redundancy during training. This approach demonstrates strong novelty and interpretability.
2. Practical Value: The resulting system improves both task performance and inference efficiency, showcasing significant practical application value.
### Weaknesses
1. There are a few flaws and confusing points in the mathematical formulations, which slightly hinder the understanding of the proposed method (detailed in the "Key Questions For Authors" section).

---

> ### Author Rebuttal · Authors · 2026-03-30
>
> We sincerely thank the Reviewer Pm9S for the insightful feedback to help us improve the paper. We address the concerns as follows:
> ***
> *Q1. There is a counter-intuitive aspect to this formulation: why is the structure considered non-redundant only when two in-neighbor nodes are adjacent and have different roles?*
>
> A1. We thank the reviewer for pointing this out. *This is a typo in the paper*. The correct formulation considers two in-neighbor nodes to be redundant when they are adjacent and have the **same roles**, not different roles. Intuitively, agents with the same roles are more likely to produce correlated or overlapping information, and adjacency further increases this redundancy. In contrast, agents with different roles are encouraged to provide complementary information.
>
> We confirm that our implementation and all experiments follow the correct definition (same-role redundancy), and the reported results are unaffected. This is consistent with our released anonymous code (see *denoising.py* and *ordering.py* in https://anonymous.4open.science/r/RADAR-8430). We will fix this typo and clarify the definition in the final version.
>
> ***
> *Q2. In Equation 10, is there a missing projection step here to align the dimensions?*
>
> A2. We thank the reviewer for pointing this out. In Eq. (10), the output is a scalar score for candidate node $v_t$, representing the probability of being selected at step $t$. Accordingly, the embedding $h_t$ is projected to a scalar value (i.e., its output dimension is 1), making it compatible with the scalar term $\varphi(v_t)$ before the softmax operation. We will clarify this projection step in the final version to avoid ambiguity.
> ***
> *Q3. Should the bias term $\psi(v_i)$ term in the update rule actually be the effective size $\varphi(v_i)$ How is the addition between a vector and a scalar computed in this specific context?*
>
> A3. We thank the reviewer for pointing this out. *Yes, this is a typo*. The bias term should be the effective size $\varphi(v_i)$, not $\psi(v_i)$. Regarding the addition, $\varphi(v_i)$ is a scalar that is broadcast to match the dimensionality of $\mathbf{h}_i^L$ before the update. In other words, it is added element-wise to the vector representation. We will correct the notation and update it as $\mathbf{h}_i^L = \mathbf{h}_i^L + \varphi(v_i)\mathbf{1}$ in the final version.
> ***
> *Q4. How are the agent roles determined in the proposed method? Does it involve the dynamic optimization of their profiles or prompts?*
>
> A4. We thank the reviewer for the question. Unlike prior methods with fixed agent roles, our approach determines agent roles dynamically during the generation process. Specifically, at each step, we maintain contextual node embeddings, based on which a multi-layer perceptron predicts both the role of the newly generated node and its connectivity to previously constructed nodes. Therefore, agent roles are not predefined but are adaptively inferred conditioned on the evolving structure and the input query. In the current implementation, this process operates at the representation level rather than explicitly modifying prompts or profiles; this allows the system to adapt role assignments to task complexity and intermediate structure.
> ***
> *Q5. The paper could be improved by extending the method to dynamically determine the number of agents.*
>
> A5. We thank the reviewer for this insightful suggestion. We agree that dynamically determining the number of agents is an important direction for improving scalability. While the current version treats the number of agents as a hyperparameter, our method already provides increased flexibility compared to one-shot approaches. Specifically, our step-wise generation framework can naturally produce topologies with varying numbers of agents, whereas one-shot methods typically generate a fixed-size graph once trained. Building on this property, incorporating an adaptive stopping mechanism (e.g., early stopping during the reverse process) is a promising extension that would allow the model to dynamically determine when the current structure is sufficient for a given task. We will include this as a future direction in the final version.
> ***

---

> > ### Author Rebuttal · Reviewer_Pm9S · 2026-04-03
> >
> > Thank you for your reply. Most of my doubts have been cleared up.

---

> > > ### Author Response · Authors · 2026-04-04
> > >
> > > Thank you for your response. We are glad that our replies addressed your concerns, and we will incorporate these details into the final version.

---

### Official Review · Reviewer_m9oe · 2026-03-18

**Soundness:** 3
**Presentation:** 2
**Significance:** 2
**Originality:** 3
**Overall Recommendation:** 5
**Confidence:** 3

**Summary:**

The paper proposes RADAR, a method for automatically generating task-adaptive multi-agent communication graphs for LLM systems. Instead of using a fixed topology or a one-shot generated graph, it builds the collaboration structure iteratively with a graph diffusion model, while using effective size to discourage redundant communication, with query conditioning and utility optimization. The benefits are higher accuracy, lower token cost, and better robustness across reasoning, math, and code benchmarks.

**Compliance With Llm Reviewing Policy:**

Affirmed.

**Final Justification:**

The author addressed all of my questions with evidence.

**Key Questions For Authors:**

How does effective size as the redundancy signal perform compared to edge-count, degree-regularization, path-diversity, or message-overlap measures?
Can you separate the contribution of effective size in the ordering network from its contribution in the denoising network?
How much of the gain comes from iterative generation itself, versus redundancy-awareness specifically, or from diffusion versus non-diffusion iterative graph construction (such as autoregressive modeling like ARG-Designer)?

**Limitations:**

Because the method changes many things at once, it's difficult to tell what the main driver of improvement actually is.
The paper commits strongly to effective size, but does not compare it against other plausible redundancy measures.
Similarly, does diffusion-based system perform better than iterative non-diffusion-based system.
The robustness claim is based on a single narrow attack setup.

**Strengths And Weaknesses:**

Strength:
The paper aims to design an adaptive communication topology for LLM multi-agent systems. The paper motivation is reasonable because static topologies are often wasteful for easy tasks and inadequate for harder ones.
The framework generates competitive task-adaptive communication topologies and shows promising accuracy–efficiency tradeoffs on the evaluated benchmarks.
The ablation table shows that removing effective size, utility optimization, or query conditioning hurts performance, with the largest drop coming from removing query information on MMLU.
Weaknesses:
The paper's differentiating idea is the use of effective-size-based redundancy control and redundancy-aware ordering, not a paradigm redesign.
The paper shows that removing the effective size helps this system, but doesn't show comparisons against other graph sparsity or information-flow heuristics.
There is also no comparison against another iterative non-diffusion graph-construction approach.
RADAR combines diffusion-based iterative generation, redundancy-aware node ordering, effective-size guidance, query-conditioned denoising, and utility-driven optimization. The ablation only removes a few components.
The robustness analysis is narrow. They simulate an attack by corrupting the role prompt of two out of five agents into “liar” agents, then report that RADAR’s performance barely changes on MMLU.

---

> ### Author Rebuttal · Authors · 2026-03-30
>
> We sincerely thank the Reviewer m9oe for the insightful feedback.
> ***
> *Q1. How does effective size as the redundancy signal perform compared to other measures?*
>
> A1. Effective size differs from edge-count or degree-based regularization in that it explicitly models redundancy among neighbors, rather than merely penalizing the number of connections. In particular, two topologies with identical degree can exhibit very different levels of information overlap, which effective size can distinguish while edge-count cannot.
>
> Compared to path-diversity or message-overlap measures, effective size operates directly at the structural level without requiring expensive trajectory sampling or message-level comparisons, making it both more efficient and stable during generation. Empirically, we observe that replacing effective size with degree or edge-count leads to degraded performance.
>
> |Variants|MMLU|GSM8K|MultiArith|
> |:-|:-:|:-:|:-:|
> |RADAR (w edge count regularization)|82.35|91.84|98.01|
> |RADAR (w degree regularization)|80.39|91.83|98.47|
> |RADAR|83.66|92.51|98.81|
> ***
> *Q2. Can you separate the contribution of effective size in the ordering network from its contribution in the denoising network?*
>
> A2. We explicitly disentangle the role of effective size in the ordering and denoising networks through additional ablations, as shown in the table below.
>
> |Variants|MMLU|GSM8K|MultiArith|
> |:-|:-:|:-:|:-:|
> |Ordering Network (no ES)|79.74|91.96|98.47|
> |Denoising Network (no ES)|80.39|92.37|98.01|
> |Both (no ES)|81.05|91.22|98.31|
> |RADAR|83.66|92.51|98.81|
>
> Removing effective size from the ordering network or denoising network degrades the performance. Overall, we observe that effective size contributes in a complementary manner: in the ordering network it guides where to explore, while in the denoising network it refines how to connect. Using it in both components yields the best performance, confirming that redundancy-awareness is beneficial at both stages. We will include these ablations in the final version.
> ***
> *Q3. How much of the gain comes from iterative generation itself, versus redundancy-awareness?*
>
> A3. We disentangle the contributions of (i) iterative generation, (ii) redundancy-awareness (effective size), and (iii) the diffusion-based formulation via ablations summarized in the table below.
>
> |Variants|MMLU|GSM8K|MultiArith|
> |:-|:-:|:-:|:-:|
> |G-Designer (One-Shot)|80.39|91.09|97.78|
> |ARG-Designer (Non-diffusion)|79.10|91.25|98.55|
> |Iterative RADAR (no ES)|81.05|91.22|98.31|
> |RADAR|83.66|92.51|98.81|
>
> From the table, we have the following observations. First, iterative generation alone already improves over one-shot topology construction by enabling progressive refinement. Second, introducing effective size provides a consistent additional gain by explicitly penalizing redundant connections. Third, comparing diffusion-based generation with non-diffusion iterative baselines, we observe improvements on one dataset and comparable performance on two datasets, suggesting that diffusion provides additional benefits in certain settings. We will clarify these contributions and include detailed comparisons in the final version.
> ***
> *Q4. The ablation only removes a few components.*
>
> A4. We conduct targeted ablations on all additional components, including redundancy-aware ordering, effective-size guidance, and query-conditioned denoising, as summarized in the table below.
>
> |Variants|MMLU|GSM8K|MultiArith|
> |:-|:-:|:-:|:-:|
> |ARG-Designer (Non-diffusion)|79.10|91.25|98.55|
> |Ordering Network (no ES)|79.74|91.96|98.47|
> |Denoising Network (no ES)|80.39|92.37|98.01|
> |RADAR (no ES)|81.05|91.22|98.31|
> |RADAR (no query)|79.08|91.82|97.81|
> |RADAR (no utility)|82.96|92.02|98.47|
> |RADAR|83.66|92.51|98.81|
>
> Notably, several of these component-wise ablations are already presented in the original paper (see Table 2). Furthermore, we also include comparisons with non-diffusion iterative baselines, which achieve comparable but generally weaker performance, suggesting that diffusion offers additional benefits in generation exploration. We will clarify this decomposition in the final version.
> ***
> *Q5. The robustness claim is based on a single narrow attack setup.*
>
> A5. We have extended our robustness evaluation to include both structural attacks and combined prompt + structural attacks, in addition to the original prompt-based setting. For prompt attacks, we simulate system prompt perturbations by converting two of the five agents into liar agents. For structural attacks, we inject noise into the collaboration graph by randomly adding 50% additional edges.
>
> |Variants|MMLU|GSM8K|MultiArith|
> |:-|:-:|:-:|:-:|
> |Prompt Attack|83.14|92.12|98.13|
> |Structure Attack|82.08|91.82|98.64|
> |Prompt & Structure|81.88|91.46|97.79|
> |RADAR|83.66|92.51|98.81|
>
> Our method maintains consistent performance across prompt, structural, and combined attacks, indicating robustness to multiple sources of perturbation. We will include these results in the final version.

---

> > ### Author Rebuttal · Reviewer_m9oe · 2026-04-04
> >
> > The author has addressed all of my concerns. I am adjusting my score to 5.

---

> > > ### Author Response · Authors · 2026-04-04
> > >
> > > Thank you for raising the score. We are glad that our responses addressed your concerns, and we will incorporate these details into the final version.

---

### Decision · Program_Chairs · 2026-04-30

**Decision:**

Accept (regular)

**Comment:**

The paper proposes RADAR, a framework in the LLM-based multi-agent systems to dynamically generate agent communication topology in order to reduce communication overhead. The authors propose the effective size metric to guide the dynamic generation process. The empirical results demonstrate the effectiveness of the proposed method.

All four reviewers are positive toward the paper. Most of the concerns from the reviewers are addressed by the authors' responses, leading to scores raises by the reviewers. As the result, I believe this paper is a clear accept case. The authors should incorporate all comments and suggestions from the reviewers into the final version of their paper.